# Validation Study for Non-Invasive Prediction of *IDH* Mutation Status in Patients with Glioma Using In Vivo ^1^H-Magnetic Resonance Spectroscopy and Machine Learning

**DOI:** 10.3390/cancers14112762

**Published:** 2022-06-02

**Authors:** Elisabeth Bumes, Claudia Fellner, Franz A. Fellner, Karin Fleischanderl, Martina Häckl, Stefan Lenz, Ralf Linker, Tim Mirus, Peter J. Oefner, Christian Paar, Martin Andreas Proescholdt, Markus J. Riemenschneider, Katharina Rosengarth, Serge Weis, Christina Wendl, Sibylle Wimmer, Peter Hau, Wolfram Gronwald, Markus Hutterer

**Affiliations:** 1Department of Neurology and Wilhelm Sander-NeuroOncology Unit, Regensburg University Hospital, 93055 Regensburg, Germany; ralf.linker@ukr.de (R.L.); peter.hau@ukr.de (P.H.); markus.hutterer@ukr.de (M.H.); 2Department of Radiology and Division of Neuroradiology, Regensburg University Hospital, 93055 Regensburg, Germany; claudia.fellner@ukr.de (C.F.); christina.wendl@ukr.de (C.W.); 3Central Institute of Radiology, Kepler University Hospital, 4021 Linz, Austria; franz.fellner@kepleruniklinikum.at; 4Division of Molecular Pathology, Neuromed Campus, Kepler University Hospital, 4020 Linz, Austria; karin.fleischanderl@kepleruniklinikum.at (K.F.); lenz.stefan@kepleruniklinikum.at (S.L.); 5Institute of Functional Genomics, University of Regensburg, 93053 Regensburg, Germany; martina195@web.de (M.H.); tim.mirus@stud.uni-regensburg.de (T.M.); peter.oefner@klinik.uni-regensburg.de (P.J.O.); wolfram.gronwald@klinik.uni-regensburg.de (W.G.); 6Institute of Laboratory Medicine, Kepler University Hospital, 4021 Linz, Austria; christian.paar@kepleruniklinikum.at; 7Department of Neurosurgery, Regensburg University Hospital, 93053 Regensburg, Germany; martin.proescholdt@ukr.de (M.A.P.); katharina.rosengarth@ukr.de (K.R.); 8Department of Neuropathology, Regensburg University Hospital, 93053 Regensburg, Germany; markus.riemenschneider@ukr.de; 9Division of Neuropathology, Neuromed Campus, Kepler University Hospital, 4020 Linz, Austria; serge.weis@kepleruniklinikum.at; 10Institute of Neuroradiology, Neuromed Campus, Kepler University Hospital, 4020 Linz, Austria; sibylle.wimmer@kepleruniklinikum.at; 11Department of Neurology with Acute Geriatrics, Saint John of God Hospital Linz, 4021 Linz, Austria

**Keywords:** glioma, *IDH* mutation, ^1^H-MRS, 2-hydroxyglutarate, linear support vector machine, independent validation

## Abstract

**Simple Summary:**

The enzyme *isocitrate dehydrogenase* (*IDH*) affects glioma cell metabolism in multiple ways. Mutation of *IDH* is not only indicative of the presence of astrocytoma or oligodendroglioma but it also comes with a better prognosis and constitutes a promising therapeutic target. Therefore, determination of *IDH* mutation status is essential in clinical practice. In most patients, tissue can be obtained by resection or biopsy to determine *IDH* status histologically. However, in some cases, this is not possible for technical reasons. We recently showed in a small cohort of patients that non-invasive determination of *IDH* mutation status using proton magnetic resonance spectroscopy (^1^H-MRS) at 3.0 Tesla (T) together with machine learning techniques is feasible in a standard clinical setting and with acceptable effort. Here, we demonstrate that our approach showed comparably good results in sensitivity (82.6%) and specificity (72.7%) in a larger validation cohort employing ^1^H-MRS at 1.5 T in a retrospective, distinct setting. We concluded that our method works well regardless of the magnetic field strength and scanner used, and thus, may improve patient care.

**Abstract:**

The *isocitrate dehydrogenase* (*IDH*) mutation status is an indispensable prerequisite for diagnosis of glioma (astrocytoma and oligodendroglioma) according to the WHO classification of brain tumors 2021 and is a potential therapeutic target. Usually, immunohistochemistry followed by sequencing of tumor tissue is performed for this purpose. In clinical routine, however, non-invasive determination of *IDH* mutation status is desirable in cases where tumor biopsy is not possible and for monitoring neuro-oncological therapies. In a previous publication, we presented reliable prediction of *IDH* mutation status employing proton magnetic resonance spectroscopy (^1^H-MRS) on a 3.0 Tesla (T) scanner and machine learning in a prospective cohort of 34 glioma patients. Here, we validated this approach in an independent cohort of 67 patients, for which ^1^H-MR spectra were acquired at 1.5 T between 2002 and 2007, using the same data analysis approach. Despite different technical conditions, a sensitivity of 82.6% (95% CI, 61.2–95.1%) and a specificity of 72.7% (95% CI, 57.2–85.0%) could be achieved. We concluded that our ^1^H-MRS based approach can be established in a routine clinical setting with affordable effort and time, independent of technical conditions employed. Therefore, the method provides a non-invasive tool for determining *IDH* status that is well-applicable in an everyday clinical setting.

## 1. Introduction

Gliomas are among the most common primary brain tumors in adults [1] and are mostly not curable. Despite a range of treatment options, such as tumor resection, radiotherapy, systemic therapies (e.g., alkylating agents), experimental targeted therapies, and tumor treating fields, the prognosis is generally poor [2,3,4].

According to the WHO classification of brain tumors 2021, the essential discriminating feature of all gliomas is the *isocitrate dehydrogenase* (*IDH*) mutation status [5]. The presence of an *IDH* mutated (*IDH*mut) status indicates a CNS WHO grade 2–4 astrocytoma or oligodendroglioma, whereas *IDH* wildtype (*IDH*wt) status in most cases implies the diagnosis of glioblastoma CNS WHO grade 4. Therefore, *IDH* status has a high value for the classification of gliomas. In addition, it is highly relevant for prognostication. While patients suffering from *IDH*wt glioblastoma receiving standard treatment have an average overall survival of 15–18 months [6,7], average overall survival can be up to 14.7 years in patients with oligodendroglioma in the presence of an *IDH* mutation and combined loss of heterozygosity (LOH) 1p/19q under sufficient treatment [8,9]. In addition, *IDH*mut is a putative, specific treatment target and has been investigated in several clinical trials using peptide vaccination and small-molecule inhibitor approaches [10,11]. Finally, repetitive evaluation of *IDH*mut might be a valuable tool for monitoring tumor response and relapse. However, sequential biopsy is not feasible in patients with glioma.

Non-invasive serial determination of *IDH* status by proton magnetic resonance spectroscopy (^1^H-MRS) can, therefore, be a valuable additional tool, supplementing standard immunohistochemical diagnosis for *IDH* status and treatment monitoring [3].

For these reasons, much effort has been put in the development of imaging methods that can reliably predict *IDH* mutation status in a clinically feasible manner. One focus is on the development of specialized applications of proton magnetic resonance imaging (^1^H-MRI) [12]. Good prediction values are reported in the analysis of standard ^1^H-MRI sequences using convoluted neuronal networks (CNNs). Employing regional cerebral blood volume (rCBV) maps, an accuracy of 83% in predicting *IDH* status in glioblastoma was obtained [13]. However, processing of radiomic features using artificial intelligence and machine learning techniques often ends up with inconclusive results, requires high technical expertise, and is time consuming in manual preparation [14,15].

In contrast, ^1^H-MRS is a well-established method to gain information about the metabolic composition of a brain lesion and is also suitable for detection of the *IDH*mut-derived metabolite 2-hydroxyglutarate (2-HG) at 2.24 ppm [16]. Current applications of ^1^H-MRS for the 2-HG based prediction of the *IDH* mutation status achieve a pooled sensitivity and specificity of 95% and 91%, respectively [16]. However, due to personnel, technical, and time requirements, these procedures are usually implemented only in a research setting. Usually, scanners with 3.0 Tesla (T) or 7.0 T and long examination times, which must be tolerated by the patient, are required [17,18]. This also means a high personnel effort both in acquisition and evaluation of the spectra. Furthermore, detection of the 2-HG peak at 2.24 ppm using a 3.0 T ^1^H-MRS is complicated by overlap with other metabolites, such as glutamine and glutamate or lipids [19]. These issues may be resolved with high technical effort such as two-dimensional measurements [20,21] or analysis of several metabolites, e.g., 2-HG and glutamate [22].

Oligodendroglioma and most astrocytoma belong to the group of *IDH*mut glioma and therefore show the signals of 2-HG in their spectra. Furthermore, LOH 1p/19q is a characteristic signature of oligodendroglioma that generally does not differ markedly in metabolism from *IDH*mut, non-co-deleted astrocytoma, except for elevated cystathionine levels that may be helpful in distinguishing *IDH* mutated gliomas with and without LOH 1p/19q [23]. Generally, ^1^H-MRS data for oligodendroglioma are sparse, but also suggest that ^1^H-MRS may be helpful in grading [24].

In our previous prospective study cohort (original cohort), we focused on a full spectrum analysis of in vivo ^1^H-MRS at 3.0 T in combination with a support vector machine (SVM) in 34 glioma patients. With this advanced diagnostic approach, we were able to achieve a specificity of 75.0% and a sensitivity of 95.5% for non-invasive determination of *IDH* mutation status [25]. This method demonstrated good technical feasibility and was well-tolerated by the patients with an additional examination period of approximately 20 min, making its use feasible in a standard clinical setting.

For further validation, we applied our approach of a full spectrum analysis in the present study to a larger retrospective cohort of patients with gliomas assessed by ^1^H-MRS at 1.5 T (validation cohort). The aim of this study was to investigate whether our established approach can be transferred to another environment with comparable specificity and sensitivity.

## 2. Materials and Methods

### 2.1. Validation Cohort

For this retrospective analysis, 100 preoperative spectra of patients with suspected glioma were available. ^1^H-MRS had been performed at 1.5 T at a single neuro-oncological site (Neuromed Campus, Kepler University Hospital, Linz, Austria) between 2002 and 2007. Spectra were excluded from further analysis if no tumor was diagnosed (*n =* 1) or *IDH* status could not be determined (*n =* 6). In these six cases, no *IDH* status was known beforehand, and no tissue was available for neuropathological reassessment. Additionally, quality of spectra was insufficient in *n =* 26 (Figure 1).

The mandatory ^1^H-MRS quality requirements were predefined for the amplitudes of the following metabolites: choline (Cho) > 0.2, creatine (Cr) > 0.1, myo-inositol (M-Ins) > 0, and creatine 2 (Cr2) > 0. Note that amplitudes are given in arbitrary units as defined by the vendor, and therefore, thresholds of amplitudes might not be applicable to other cohorts. Details and examples of sufficient and insufficient spectrum quality are shown in the supplement (Appendix A). Ultimately, 67 spectra were included from 44 patients with *IDH*wt and 23 patients with *IDH*mut status.

For 32 samples, *IDH* status had been assessed previously in clinical routine diagnostics by immunohistochemistry for the most common *IDH* mutation *IDH1 R132H* [26]. Neuropathological re-evaluation with next-generation sequencing (NGS) for this study was performed in 35 cases described in detail in Section 2.2. Thirteen patients with immunohistochemically confirmed *IDH*wt status were less than 55 years old at initial diagnosis. According to the current diagnostic guidelines [5,27], additional *IDH* sequencing was performed to detect rare *IDH* mutations particularly in these cases. In nine samples, *IDH*wt status was confirmed, whereas in four cases, DNA sequencing was not possible (no tissue available: *n =* 3, DNA concentration too low: *n =* 1).

From these four patients, three had a predefined, histologically confirmed glioblastoma, WHO grade IV, without notice of a lower-grade precursor lesion. Thus, these cases could be most likely considered primary *IDH*wt glioblastomas. The remaining sample was a histologically classical malignant diffusely infiltrating astrocytic glioma, most likely corresponding to a molecularly defined *IDH*wt glioblastoma according to the WHO 2021 classification.

### 2.2. Neuropathological Assessment of Glioma Tissue

#### 2.2.1. Immunohistochemistry

Immunohistochemistry was performed on formalin-fixed and paraffin-embedded (FFPE) 5μM thick sections on Superfrost Plus Adhesion Microscope slides (J1800AMN2, Epredia, Kalamazoo, MI, USA). Deparaffinized, rehydrated sections underwent antigen retrieval in the PT Module Lab Vision (Thermo Scientific, Loughborough, UK) using the Dewax and HIER Buffer L (pH 6) (Epredia, DA Breda, The Netherlands). All subsequent steps were carried out using the Lab Vision Autostainer 480S (Thermo Scientific, Loughborough, UK) and the Ultravision Quanto Detection System HRP (TL-125-QHL, Epredia, DA Breda, The Netherlands). The reaction product was visualized using diaminobenzidine chromogen (DAB Quanto, TA-125-QDX, Epredia, DA Breda, The Netherlands). Sections were counterstained with Mayer’s hematoxylin (Thermo Scientific, Loughborough, UK). The following primary antibody was used: anti-*IDH1* R132H (dilution 1:50, mouse monoclonal, clone H09, Dianova, Hamburg, Germany). As negative control, the primary antibody was omitted and was replaced by normal mouse serum.

#### 2.2.2. DNA Extraction and Quantification

DNA extraction from FFPE tissue was performed with the blackPREP FFPE DNA Kit (Analytik Jena, Jena, Germany) and quantified with a Qubit 3.0 System using the Qubit™ dsDNA HS Assay Kit (Invitrogen, Dreieich, Germany).

#### 2.2.3. Next-Generation Sequencing (NGS)

The *IDH* mutation status of samples with unclear immunohistochemistry and of patients younger than 55 years with negative *IDH* immunohistochemistry was determined by NGS using the Ion AmpliSeq™ Comprehensive Cancer Panel (Life Technologies, Darmstadt, Germany). NGS libraries were generated with the Ion AmpliSeq^TM^ Library 2.0 Kit (Life Technologies, Darmstadt, Germany, User guide MAN0006735 Rev. B.0). DNA was pretreated for 30 min with 1 unit Uracil-DNA Glycosylase (Life Technologies, Darmstadt, Germany). Template preparation, chip loading, and sequencing were performed with the Ion Chef™ and the Ion GeneStudio™ S5 Plus Systems according to the Ion 510™ & Ion 520™ & Ion 530™ Kit—Chef user guide (MAN0016854 Rev. E.0, Life Technologies, Darmstadt, Germany). Data were analyzed with Torrent Suite™ software version 5.12.2 and Ion Reporter software version 5.16. Read alignments at the *IDH1* and *IDH2* hotspots were visually inspected with the Integrative Genomics Viewer (Broad Institute, Cambridge, MA, USA).

#### 2.2.4. Multiplex Ligation-Dependent Probe Amplification (MLPA)

The copy number status of chromosomes 1p and 19q was determined by MLPA using the P088 Oligodendroglioma 1p-19q probe mix (MRC Holland, Amsterdam, Netherlands, Lot. D1-1119, COA Version 01) according to the MLPA General Protocol MDP version-007. Data were analyzed with Coffalyser.Net software version v.210604.1451 (MRC Holland, Amsterdam, Netherlands).

### 2.3. Data Acquisition of ^1^H-MRS

In the validation cohort, magnetic resonance imaging (MRI) and ^1^H-MRS were performed on clinical whole-body scanners, as performed in the original cohort. While the data for the original cohort were acquired on a state-of-the art 3.0 T scanner (MAGNETOM Skyra, Siemens Healthcare, Erlangen, Germany) between 2015 and 2019, data for the validation cohort had been measured between 2002 and 2007 on a 1.5 T scanner (MAGNETOM Symphony, Siemens Healthcare, Erlangen, Germany). MR spectra were acquired by a single-voxel technique employing a point-resolved spectroscopy (PRESS) sequence with a short echo time (TE) of 30 ms.

There were two major differences between the training and validation cohort: the voxel size for ^1^H-MRS was kept constant in the original cohort, while the voxel size was adapted to tumor size and shape in the validation cohort. In addition, positioning of the voxel into the most representative part of the tumor was guided exclusively by standard MRI for the validation cohort, while voxel positioning had been also supported by O-(2-^18^F-fluoroethyl)-L-tyrosine (^18^F-FET) positron emission tomography (PET) in the original cohort. Further technical details are given in the supplement (Appendix A) and for the original cohort in Bumes et al., 2020 [25].

### 2.4. H-MRS Data Preprocessing

For the original cohort, in each spectrum, 1024 data points were acquired over a spectral width of 1200 Hz at 3.0 T, while for the validation cohort, 1024 data points were recorded over a spectral width of 1000 Hz at 1.5 T. Further technical details are given in the supplement (Appendix A).

As the predictive model was trained on the original data, before application to the validation data (Figure 2), data sets had to be matched to each other. To this end, all spectra were aligned with respect to the choline peak while accounting for the differences in field strength and spectral width between the two data sets.

### 2.5. H-MRS Classification

First, all ^1^H-MR spectra of the original cohort were used for training of an SVM with linear kernel. To this end, the R library e1071 (http://cran.r-project.org/web/packages/e1071/(accessed on 22 February 2022)) was applied. Throughout training, the cost parameter was left on its default value of one. Next, the preprocessed and adjusted spectra of the validation cohort were presented to the trained model as test cases.

The complete data acquisition and analysis pipeline is summarized in Figure 2, and a detailed description is presented in the supplement (Appendix A).

We have developed a user-friendly R-shiny app that allows application of the trained model to additional data. New data are automatically adapted with respect to field strength, spectral width, and number of acquired data points. The app including the fully trained model predicts *IDH* mutation status and is freely accessible for scientific purposes: https://www.uni-regensburg.de/medicine/functional-genomics/staff/prof-wolfram-gronwald/software/index.html (accessed on 29 April 2022) (Appendix A). In addition, the app can also be used as a web application: https://IDH-prediction.spang-lab.de (accessed on 29 April 2022).

## 3. Results

### 3.1. Patient Characteristics of the Validation Cohort

Among the 67 patients enrolled, 44 cases were *IDH*wt, whereas 21 patients carried an *IDH1* R132H mutation (Figure 1). Additionally, two rare *IDH* mutations (R132S and R172M) were detected, bringing the total to 23 *IDH*mut patients.

Age at initial diagnosis was higher in patients with *IDH*wt glioma, with a median of 61 years (range 21 to 83 years), than in patients with *IDH*mut glioma, with a median of 37 years (range 18 to 69 years). The exact age distribution at first diagnosis is shown in the supplement (Appendix A).

As expected, patients with *IDH*wt status usually showed WHO grade IV in the initial histological assessment according to the WHO classification of brain tumors 2000 (Table 1).

In total, 30% of patients with *IDH*mut status (*n =* 7) presented with LOH 1p/19q, thus confirming an oligodendroglioma according to the WHO classifications 2016 and 2021. No LOH 1p/19q was present in ten samples (43%), while assessment was not possible in six cases (26%) due to technical limitations.

Since an *IDH*wt status excludes the diagnosis of oligodendroglioma, we did not determine LOH status in *IDH*wt samples in general.

Patient characteristics of the original cohort are published in Bumes et al. [25].

### 3.2. IDH Mutation Caused Specific Alterations in ^1^H-MR Spectra

Distinct alterations characteristic for the presence of an *IDH* mutation were apparent in the spectra beyond the 2-HG peak at 2.24 ppm, especially in the region between 4.1 ppm–3.5 ppm, as already reported for the original cohort [25]. In two exemplary spectra of the validation cohort (Figure 3A,B) acquired with a routine 1.5 T scanner and without standardized voxel sizes, this region is marked by a blue box including an increase in the region of the M-Ins signal at 3.53 ppm in the *IDH*mut spectrum (Figure 3A). In Figure 3C, the averaged spectra of the *IDH*mut and *IDH*wt group show the variation in the region of interest even more clearly.

Thus, the spectral differences observed in the original cohort held true for the validation cohort. Furthermore, clear differences between the *IDH*mut and *IDH*wt group were apparent in the spectral region corresponding to lactate and methylene (CH2) groups of lipids at around 1.3 ppm, showing a considerably increased intensity in the *IDH*wt group (Figure 3C).

### 3.3. A Linear SVM Provided High Sensitivity and Specificity in Detecting IDHmut

Next, a linear SVM was applied to the classification of ^1^H-MR spectra from the validation cohort. Employing the original cohort, the number of discriminatory features was optimized in a cross-validation for an optimal prediction accuracy, resulting in the selection of two predictive features. The final model was subsequently trained on the complete original cohort, using the two most differentially expressed features between the *IDH*wt and *IDH*mut groups. Following matching of the validation and original data that were independently acquired at 1.5 T and 3.0 T, respectively, the trained model was applied to achieve classification of the validation data in the next step, as described in the Materials and Methods section.

We showed an accuracy of 76.1% (95% CI, 64.1–85.7%), a sensitivity of 82.6% (95% CI, 61.2–95.1%), and a specificity of 72.7% (95% CI, 57.2–85.0%). The area under the ROC curve (AUC) amounted to 0.82 (Figure 4). A full description of the obtained statistical parameters for both the original and the validation cohort is given in Table 2.

Visual inspection of false positive and false negative predicted spectra revealed no evidence of systematic error. Quality of the spectra was sufficient in all false predicted samples; examples are shown in the supplement (Appendix A).

### 3.4. SVM Did Not Predict LOH 1p/19q Status

Subsequently, we investigated whether the non-invasive prediction of the LOH 1p/19q status was possible. To this end, an SVM was trained to separate the spectra of *IDH*mut subgroups with and without combined LOH 1p/19q. According to the current diagnostic standards, a combined LOH 1p/19q is present only in *IDH*mut gliomas, so the neuropathological re-evaluation of the LOH status was performed only in *IDH*mut samples as described in Section 2.2.4.

The small group sizes of the validation cohort (Figure 5A), due to technical reasons caused by the age of the samples, did not allow a statistically meaningful analysis of the spectra. With the same technical assessment in both cohorts, it was feasible to add the samples from the original cohort to the validation cohort to obtain a larger group (total cohort: *IDH*mut with LOH 1p/19q, *n =* 13; *IDH*mut without LOH 1p/19q, *n =* 26).

As demonstrated in the averaged spectra (Figure 5B), only minor differences in the full spectrum analysis of the total cohort were visible between *IDH*mut subgroups with and without LOH 1p/19q. Results of the used classification approach showed that the observed differences did not allow reliable prediction of the LOH 1p/19q status (not shown).

## 4. Discussion

Here, we showed that assessment of *IDH* mutation status using single-voxel ^1^H-MRS on a 1.5 T scanner produced acceptable sensitivity (82.6%), specificity (72.7%), and AUC (0.82), if compared to tissue-based methods. These findings corroborate data from our recently published original cohort, which yielded a higher sensitivity (95.5%) but similar specificity (75.0%) and AUC (0.83).

The data provided in Table 2 show that for both cohorts, reliable classification results were obtained. This is especially noteworthy as disease prevalence differed considerably between the two cohorts. In the original cohort used for training of the model, 22 out of 34 patients were bearing an *IDH* mutation (65%), while in the validation, this was true for only 23 out of 67 patients (34%). This further demonstrates the robustness of the proposed approach.

The good reproducibility is further remarkable given the different technical conditions. The single-voxel spectroscopy in our validation cohort, which used spectra generated from 2002 to 2007, was performed on a 1.5 T scanner, while the original cohort was examined on a modern 3.0 T scanner from 2015 to 2020. Although similar measurement parameters were used in both cohorts, the ^1^H-MRS at 1.5 T yielded, compared to 3.0 T, a reduced spectral resolution and a lower signal-to-noise ratio (SNR). Therefore, the technical conditions of the validation cohort were inferior to those of the original cohort.

Furthermore, differences in the positioning and configuration of the voxel were evident for both cohorts. In our original cohort, ^18^F-FET PET was used as an additional tool for voxel positioning, which was not available for the validation cohort. Therefore, voxel positioning in the validation cohort might not always have captured the region of highest biological activity, further impeding sample classification.

While in the original cohort, a constant isotropic voxel length of 14 mm was used to obtain comparable spectra, the voxel size in the validation cohort was adjusted to the region of interest (based on structural MRI) to achieve the best possible coverage of the tumor region. This approach causes minimal partial volume effects from surrounding tissue. However, for small voxel size, the number of averages was not increased due to time reasons, resulting in a relatively low SNR in some cases. To exclude spectra with too low SNR, quality criteria for amplitudes of metabolites were introduced (Figure 1 and Appendix A). On the other hand, due to the use of a constant isotropic voxel length of 14 mm, there was a higher risk of partial volume effects in the original cohort.

Despite these substantial differences, our approach of full spectrum analysis proved to be sufficiently robust to assess *IDH* mutation status. Hence, the approach can be applied easily to other data sets acquired with different technical settings. The technical requirements for determination of *IDH* mutation status were also met with 1.5 T scanners, which are frequently used in clinical practice. Furthermore, the additional examination time of approximately 20 min is tolerable for most patients. Note that even better results may be expected with the use of 3.0 T scanners, which are becoming increasingly available in clinical routine.

A meta-analysis of 14 studies comprising 460 glioma patients showed excellent diagnostic performance of 2-HG ^1^H-MRS for prediction of *IDH*mut glioma, with a pooled sensitivity and specificity of 95% (95% CI, 85–98%) and 91% (95% CI, 83–96%), respectively [16]. However, determination of the 2-HG peak is technically much more elaborate and difficult to implement in clinical routine. Usually, a scanner with at least 3.0 T, often 7.0 T, is used with long examination times that are not tolerable for neurologically impaired patients. Berrington et al. [17] showed that reliable detection of 2-HG is possible using an ultra-high field 7.0 T scanner. Similarly, Shen et al. [18] employed a 7.0 T scanner for the detection of 2-HG. Additionally, two-dimensional MR spectra have been successfully applied for the reliable detection of 2-HG [20,21].

Other approaches, such as the use of radiomic features, employ standard MRI sequences and therefore skip the additional examination time for ^1^H-MRS. However, the prediction quality depends very much on the algorithm used [14]. Furthermore, it is a technically complex and time-consuming process (e.g., tumor segmentation is needed) difficult to be implemented in clinical routine.

In comparison, despite its simplicity and lower technical requirements, our approach using full spectrum analysis still allowed reliable prediction of *IDH* mutation status with both good sensitivity and specificity. We encourage colleagues to use our free app to predict *IDH* mutation status for scientific purposes (https://www.uni-regensburg.de/medicine/functional-genomics/staff/prof-wolfram-gronwald/software/index.html (accessed on 29 April 2022)) (Appendix A), which is also available as a web application (https://IDH-prediction.spang-lab.de (accessed on 29 April 2022)).

Published data using a MEGA-PRESS sequence on a 3.0 T scanner showed that the spectra of *IDH* mutant gliomas with and without LOH 1p/19q are distinct, especially in cystathionine levels [23]. We could not replicate these alterations in our analysis, probably because of different technical conditions, as we used a PRESS sequence on a 1.5 T scanner. Furthermore, only a small number of histologically confirmed spectra with LOH 1p/19q (*n =* 13) could be included in our analysis.

This study has several weaknesses. First, both the original and the validation cohort are relatively small. We had to exclude some spectra of the validation cohort either due to unavailability of *IDH* status or poor spectra quality.

Neuropathological reassessment of *IDH* status by sequencing was not possible for the entire validation cohort. Previously, *IDH* status was determined by immunohistochemistry. Here, if feasible, *IDH* status was confirmed by sequencing, particularly in patients with *IDH*wt status who were younger than 55 years at initial diagnosis. However, due to the time passed, tissue was not always available or technical problems prevented neuropathological reassessment. We believe the risk of missing a rare *IDH* mutation in patients with *IDH*wt immunohistochemistry but no retroactive sequencing to be very low. As outlined above, the clinical information and/or the histological pattern in these cases were consistent with *IDH*wt glioblastoma and no suspicion of misclassification was found in the analysis of the spectra.

This study also has several strengths. Despite the different conditions used to acquire ^1^H-MR spectra in the two cohorts, diagnostic performance of our approach, in particular with regard to specificity, was sufficiently robust to predict *IDH* mutation status with acceptable effort of time and human resources.

## 5. Conclusions

We conclude that assessment of *IDH* mutation status is feasible in a standard clinical setting with little time and staff effort using ^1^H-MRS on routine 1.5 T and 3.0 T MR scanners. The method may, therefore, be of value for the assessment of *IDH* status in patients where biopsy or resection is not feasible, and in a setting where sequential performance is required. Thus, our approach works independently of various technical requirements and should be further developed to enable the non-invasive determination of *IDH* mutation status in clinical practice.

## Figures and Tables

**Figure 1 cancers-14-02762-f001:**
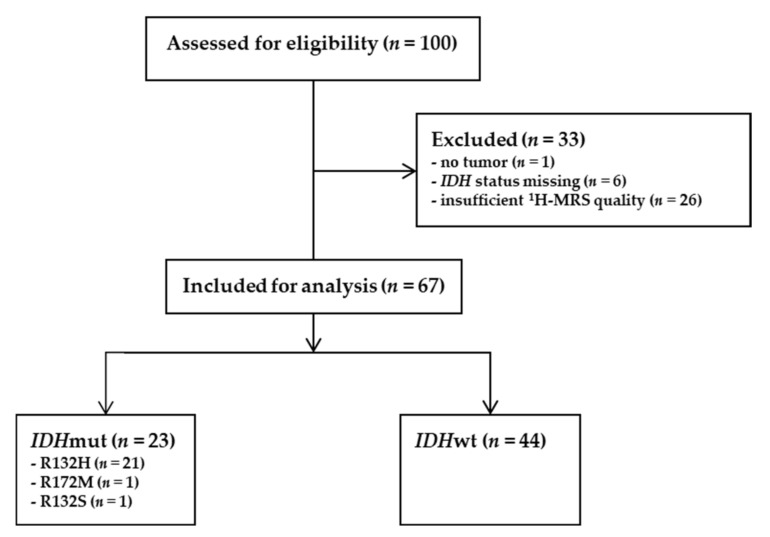
Overview of exclusion criteria and mutation status of ^1^H-MR spectra. ^1^H-MRS: proton magnetic resonance spectroscopy; *IDH: isocitrate dehydrogenase*; mut: mutated; wt: wildtype.

**Figure 2 cancers-14-02762-f002:**
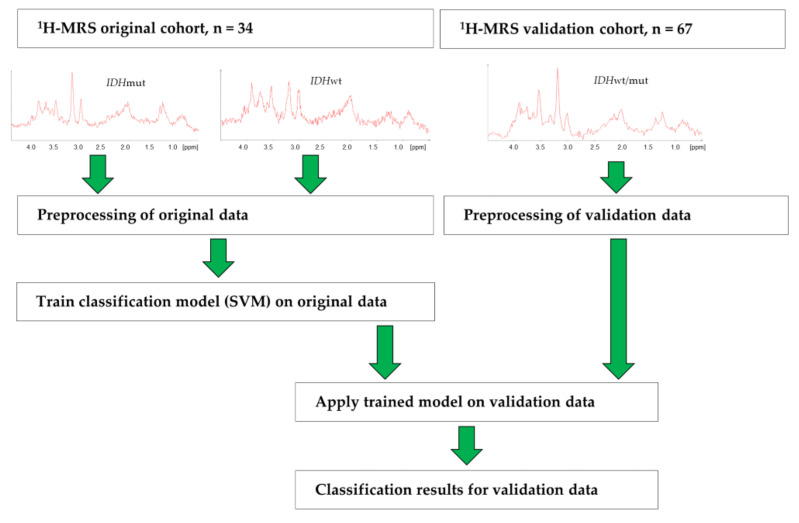
Workflow of data collection and analysis in the original and validation cohort. ^1^H-MRS: ^1^H-magnetic resonance spectra; *IDH*: *isocitrate dehydrogenase*; mut: mutated; wt: wildtype; SVM: support vector machine.

**Figure 3 cancers-14-02762-f003:**
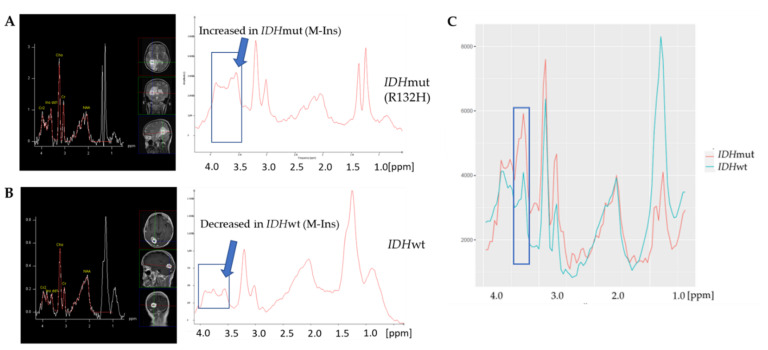
^1^H-MR spectra obtained for an *IDH*mut patient with R132H mutation (**A**) and an *IDH*wt patient (**B**) of the validation cohort. (**C**) Averaged spectra of *IDH*mut group (marked in red, *n =* 23) and of *IDH*wt group (marked in green, *n =* 44) of the validation cohort. The region used for classification including M-Ins peak is indicated by the blue box. ^1^H-MR: ^1^H-magnetic resonance; *IDH*: *isocitrate dehydrogenase*; mut: mutated; wt: wildtype; M-Ins: myo-inositol.

**Figure 4 cancers-14-02762-f004:**
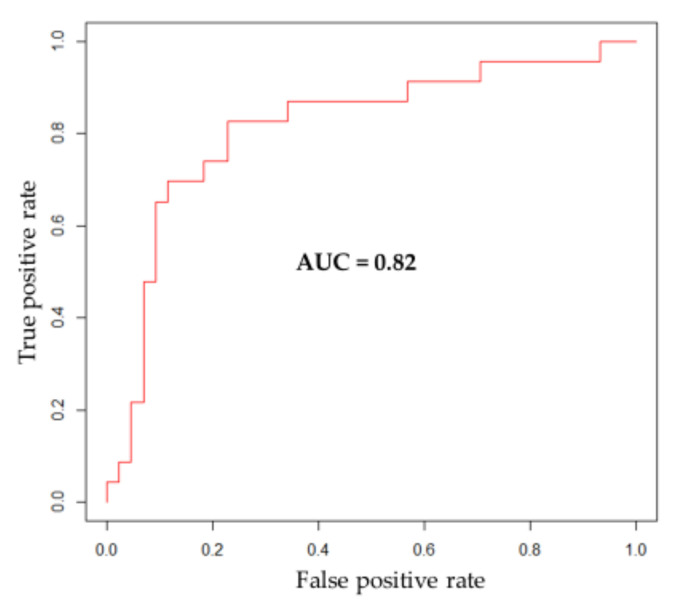
Area under the ROC curve (AUC) in prediction of *IDH* status in the validation cohort. *IDH*: *isocitrate dehydrogenase*.

**Figure 5 cancers-14-02762-f005:**
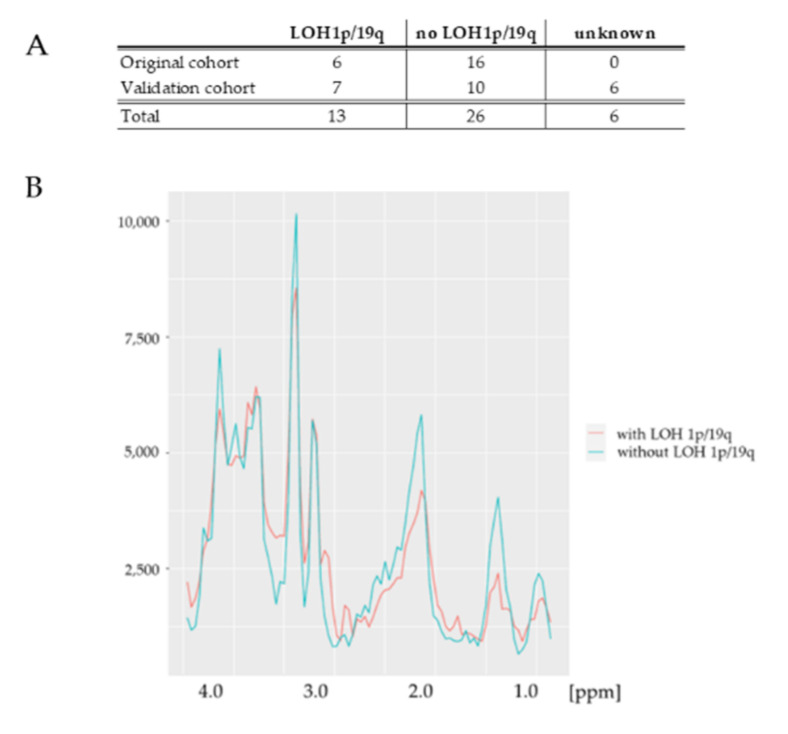
LOH 1p/19q status in the original and validation cohort in the *IDH*mut subgroup. (**A**) Distribution of LOH 1p/19q status in the original, validation, and total cohort. (**B**) Averaged spectra of *IDH*mut group with LOH 1p/19q (marked in red, *n =* 13) and of *IDH*mut group without LOH 1p/19q (marked in green, *n* = 26) in the total cohort. LOH: loss of heterozygosity; *IDH*: *isocitrate dehydrogenase;* mut: mutated.

**Table 1 cancers-14-02762-t001:** Characteristics of the validation cohort. WHO grade according to initial histology (WHO classification of brain tumors from 2000). *IDH*: *isocitrate dehydrogenase*; mut: mutated; wt: wildtype; LOH: loss of heterozygosity.

	*IDH* Mutation			
	*IDH*wt		*IDH*mut	
	Number	In %	Number	In %
Sex				
men	26	59%	15	65%
women	18	41%	8	35%
WHO				
II	3	7%	13	57%
III	4	9%	9	39%
IV	36	82%	1	4%
unknown	1	2%	0	0%
LOH1p19q				
no	4	9%	10	43%
yes	0	0%	7	30%
unknown	40	91%	6	26%
Total	44		23	

**Table 2 cancers-14-02762-t002:** Summary of classification results obtained for the original and the validation cohort. CI: confidence interval; AUC: area under the curve.

	Original Cohort	Validation Cohort
	Value (95% CI)	Value (95% CI)
Accuracy (%)	88.24 (72.55–96.70)	76.12 (64.14–85.69)
Sensitivity (%)	95.45 (77.16–99.88)	82.61 (61.22–95.05)
Specificity (%)	75.00 (42.81 94.51)	72.73 (57.21–85.04)
Positive likelihood ratio	3.82 (1.43–10.22)	3.03 (1.80–5.08)
Negative likelihood ratio	0.06 (0.01–0.42)	0.24 (0.10–0.59)
Positive predictive value (%)	87.50 (72.35–94.93)	61.29 (48.55–72.66)
Negative predictive value (%)	90.00 (56.33–98.43)	88.89 (76.32–95.20)
Disease prevalence (%)	64.71 (46.49–80.25)	34.33 (23.15–46.94)
AUC	0.83	0.82

## Data Availability

Data are contained within the article or Appendix A.

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
