# Peer review of "Validation Study for Non-Invasive Prediction of IDH Mutation Status in Patients with Glioma Using In Vivo 1H-Magnetic Resonance Spectroscopy and Machine Learning"

_cancers, 2022, doi:10.3390/cancers14112762_

Round 1
Reviewer 1 Report
The manuscript is a validation of a previously published algorithm for the detection of IDH mutation in gliomas from MRS data in a retrospective cohort of patients acquired with 1.5 T scanners.
I think this work is relevant for the clinical practice of neuro-oncologic imaging. As stated in the manuscript, the identification of IDH mutation in gliomas is crucial for the classification of these neoplasms as well as for correct treatment and prognostication. While IDH identification through conventional imaging remains elusive, MRS techniques represent a great tool to precisely detect this mutation, as demonstrated by several studies. Nevertheless, the interpretation of MRS results is not always straight-forward, and the diffusion of this technique in the clinical practice is often too limited.
Although the Authors already published a paper on the topic, I think the validation on a retrospective cohort is still relevant. The translation of AI-models to the clinical practice depends on their generalizability, as tested in validation studies. The good performance of the model in the retrospective cohort, including heterogeneous standards for voxel positioning as well as different scanners from the original population, supports model generalizability and allows future researchers to use the algorithm in different centers.
I have the following suggestions:
- The Authors claim that the accurate performance of their model was due to differences in the spectra in the frequency interval of HG. This appears logical since HG is the detected metabolite in MRS evaluation of IDH mutation. Nevertheless, I wonder if the Authors have thought of testing the prediction accuracy of their model in different frequency sub-intervals of the spectra. In this way, the Authors could demonstrate their claim by showing higher accuracy of the model in the sub-interval including the HG peak.
- I suggest to expand the discussion about AI-powered detection of IDH based on MRI. Particularly, deep-learning methods achieved good performance in previous studies [doi: 10.3390/jpm11040290]. If IDH detection could be achieved from MRI sequences which are routinely performed, there would be no need to perform MRS, reducing scanning times. However, deep-learning algorithms are not easy to implement and detection of IDH based on conventional imaging can be time-consuming due to lack of satisfactory automatic tumor-segmentation. This fact significantly limits the clinical application of these models, differently from the method presented in this study.
- MRS detection of HG is a quite sensitive and specific technique. In these terms, the accuracy achieved by the model needs to be framed in the context of the performance of routine MRS evaluation by neuroradiologists. Did the Authors consider conducting a blinded evaluation of the spectra with radiologists and compared the performance to the AI model? It would be interesting to understand how the model behaves compared to the standard evaluation by visual inspection. This analysis would test the superiority of the AI model in a clinical setting. In any case, I suggest to discuss some false positive and false negative results to provide an understanding of the model shortcomings.
- Due to the relevance of the study, I suggest to make the algorithm available on an open-source platform and provide the link to the data within the manuscript.
Finally, a few minor English corrections should be performed; I suggest reviewing the paper for typos (for example line 79, ‘into developing of’ to be corrected to ‘in the development of’)
Thank you
Reviewer 2 Report
Bumes et al. report a validation study of their previous research (Ref number 22), predicting IDH mutation status in glioma patients with MRS and machine learning. Although the findings are interesting, the following points significantly decrease the validity of the study and need to be addressed.
- Why was the validation study done on MRS performed much earlier (between 2002 and 2007) than the original cohort (between 2015 and 2019), done on a different scanner 1.5T vs 3.0T, at a different institution, with different parameters? This makes the pathological /molecular diagnosis in old tissues very difficult with many missing data. The only point I can think of is that this proves the original method can be adapted to MRS data from other institutions with moderate accuracy.
- The sensitivity and specificity of IDH mutation prediction is not high, with less specificity than that reported in the past detecting 2HG and glutamate using a clinical scanner (Nagashima et al, Neuro-Oncol, 2016).
- Has it been reported that myo-inositol is higher in IDH-mutant gliomas compared to IDH-wildtype? Many studies show non-significant differences (Natsumeda et al, Acta Neuropathol Commun, 2014; Nagashima et al, Neuro-Oncol, 2016).
- Simple summary line 30-31 states "However, analysis of tumor tissue is no always possible." when in fact biopsy of tumor tissue in glioma patients is almost always possible, excluding maybe diffuse midline gliomas located in the pons.
- A space is needed between numbers and units. Please correct.
- The authors should discuss the following paper which successfully predicts 1p/19q codeletion with high accuracy by detection of cystathionine. (Branzoli et al., Neuro-Oncol, 2019.)
Round 2
Reviewer 1 Report
The Authors addressed my concerns providing reasonable integrations to the manuscript.
Reviewer 2 Report
The authors have satisfactorily made changes based on my past comments and suggestions and is now considered suitable for publication.